# A Process-Level Method for Creativity Evaluation in LLM-Assisted Learning

## Abstract

Interpretable creativity assessment remains challenging, and the adoption of large language models (LLMs) in education amplifies issues of subjectivity and opacity. This study presents a process-level evaluation approach for LLM-assisted learning that attributes learner-versus-model contributions from multi-turn student–LLM dialogues and scores four expert-elicited dimensions with rationale texts. Using 1,273 cleaned dialogues from 81 undergraduates across multi domains, an auditable attribution protocol and an instruction-tuned evaluator are introduced to produce process-linked, interpretable rationales. Empirical evaluation with expert assessments indicates alignment with expert judgments. Claims are explicitly scoped to the studied tasks and domains, and code and evaluation scripts will be released for reproducibility.

## 1 Introduction

Systematic observation and evaluation of the dynamic emergence of creative thinking form the foundation for cognitive ability assessment and educational practice (Hennessey & Amabile, 2010). Such assessments not only facilitate the revelation of the formative mechanisms of human creative cognition (Beaty et al., 2018) but also support the cultivation of next-generation innovative capabilities within the educational system (Plucker et al., 2004). Furthermore, as creativity is a critical driver of societal progress, its reliable evaluation constitutes a core objective in the educational domain (Runco & Jaeger, 2012). Consequently, the development of interpretable measurements of creativity has long been a key focus of research in cognitive neuroscience and behavioral studies.

### 1.1 Inherent Limitations of Traditional Creativity Assessment

However, traditional methods struggle in real settings and lack suitable tools to capture thinking in situ. Two dominant paradigms illustrate this. First, standardized questionnaires such as the TTCT (Torrance Tests of Creative Thinking) compress creativity into a single score from four dimensions (fluency, flexibility, originality, elaboration) (Torrance, 1974). In tasks like alternative uses, counting ideas measures output but misses the cognitive leaps—from "wrapping" to "crafting" to "fuel" (Guilford, 1967; Silvia et al., 2008). Second, lab-based protocols (e.g., think-aloud with recorded transcripts) trace associations via semantic/temporal analyses, yet remain test-like: participants know they are being evaluated, and the resulting traces diverge from authentic problem-solving (Ericsson & Simon, 1993; Mednick, 1962; Kenett et al., 2014). These decontextualized approaches reveal a core pre-LLM bottleneck: no tooling to capture real-time cognitive dynamics in authentic contexts (Finke et al., 1992; Beaty et al., 2018). Questionnaires offer only snapshots; lab traces rarely transfer to everyday creation (Plucker et al., 2004). As a result, traditional assessments miss sudden insights and the evolving logic of ideas in real-world scenarios (Hennessey & Amabile, 2010).

### 1.2 Exacerbation of Assessment Challenges in the Era of LLMs

The rapid uptake of LLMs in education amplifies long-standing weaknesses of outcome-focused assessments and creates a governance paradox in higher education (Kasneci et al., 2023; Hennessey & Amabile, 2010). On the one hand, curricula must embrace modern tools so students can use LLMs as scaffolds for writing and research (Wood et al., 1976). On the other, over-reliance can erode independent thinking, while instructors—lacking reliable ways to separate student-authored

from LLM-generated content—struggle to assess real ability or intervene effectively (Hennessey & Amabile, 2010; OpenAI, 2023). A common remedy—requiring revision or version histories—uses traditional means for a new problem: manual traces neither reconstruct how students co-develop ideas with LLMs nor fit LLM-mediated digital workflows (Siemens & Baker, 2012), producing a stalemate where strict control impedes adaptation and non-control invites creativity decay. Compounding this, assessment faces a dual failure: human judges are poor at identifying AI-authored or co-authored text (Zellers et al., 2019; Gehrmann et al., 2019), and automated detectors/watermarks degrade under paraphrase and other adversarial edits, making them unsuitable as the foundation for high-stakes evaluation (OpenAI, 2023; Krishna et al., 2023; Kirchenbauer et al., 2023b;a).

## 1.3 LIMITATIONS OF EXISTING RESEARCH

In light of the aforementioned challenges, existing research on creativity in the context of LLMs exhibits a significant gap. The core issue is that classical assessment criteria have become obsolete, and there is a lack of adapted frameworks suitable for these new scenarios. These "classical assessment criteria" refer precisely to the core dimensions of the Torrance Tests of Creative Thinking (TTCT) mentioned earlier: fluency, flexibility, originality, and elaboration (Torrance, 1974). These dimensions were conceived in an era devoid of AI assistance and are thus capable only of measuring an individual's ability to generate ideas independently. They entirely fail to encompass the new innovation competencies required in the age of LLMs (Kasneci et al., 2023). For example, competencies such as "interdisciplinary innovation" (e.g., using an LLM to integrate knowledge from biology and engi- neering to design a biomimetic device) and "efficiency in resource integration" (e.g., rapidly screen- ing multi-domain literature via an LLM to formulate a research approach) fall outside the evaluation scope of these classical standards (Kasneci et al., 2023). More critically, current research tends to either focus exclusively on the "novelty of the final product" (Amabile, 2011) or the "frequency of LLM use," thereby overlook- ing the "causal relationship between the process trajectory and creative ability" (Sio & Ormerod, 2009). Alternatively, even when attempts are made to track the process (e.g., through semantic analysis), they often lack a clear logic for "differentiating human versus machine contributions" (Krishna et al., 2023) and fail to establish a meaning- ful connection with classical theories of creativity (Kenett et al., 2014).

## 1.4 THE PROPOSED RESEARCH SOLUTION

To address the foregoing pain points, this paper proposes Creativity–Reality Evaluation with Decoupled Ontology (CREDO)—a process-level, attribution-based creativity assessment framework for human–LLM collaboration. CREDO is process-evidence centric: rather than judging only the final product, it evaluates creativity around the evolution of thinking in authentic tasks. Its core capacities (e.g., interdisciplinary integration, problem reframing, risk-driven innovation, and resource-integration efficiency) remedy blind spots of traditional outcome-oriented tools in collaborative settings and remain aligned with mainstream theories in cognitive science and education (Chi & Wylie, 2014; OECD, 2024; Sternberg, 1985) (details of alignment are provided in Section 3). To operationalize the framework, we design two components. First, the Innovation Tracing Atlas (ITA)—which decomposes multi-turn "student–LLM" dialogues, turn by turn, into cognitive steps such as questioning–reframing–integrating–generating, and differentiates student-initiated operations from LLM scaffolding, thereby transforming previously invisible thinking trajectories into auditable, reusable process evidence. Second, the instruction-tuned evaluator—which applies parameter-efficient fine-tuning on a large model (Hu et al., 2021) to output 1–5 scores along CREDO's four dimensions and generate textual rationales, supporting interpretable and reviewable process-based assessment (training and inference settings are detailed in Section 4) and implemented on the DeepSeek family (DeepSeek-AI et al., 2025; DeepSeek-AI, 2025).

## 2 RELATED WORK

**Traditional Outcome-Oriented Assessment:** The assessment of human creativity has long relied on outcome-oriented standardized instruments that target final products or simplified tasks. A representative measure is the TTCT, which evaluates divergent thinking across four dimensions—fluency, flexibility, originality, and elaboration (Torrance, 1974). Common methods also include CAT (assessment of authentic works) (Amabile, 1982), AUT/RAT (divergent/convergent thinking) (Guil-

ford, 1967; Mednick, 1962), and CAQ (self-reported achievements) (Carson et al., 2005). These tools are ill-suited to the need for process evidence under human–LLM interaction (Kasneci et al., 2023; Siemens & Baker, 2012).

**Related Studies and Their Limitations:** Existing cross-cutting research falls broadly into three strands: evaluating LLMs themselves (adapting AUT/TTCT to measure the model rather than the learner); LLM-as-a-Judge (using LLMs to score final products, but scores are sensitive to prompt/style biases and lack an auditable causal evidence chain) (Zheng et al., 2023; Li et al., 2023); and analyses of human–AI co-created outputs (showing average quality gains alongside style convergence). A common limitation across all three is a continued focus on outcomes, with insufficient attention to process trajectories and human–machine attribution (Zellers et al., 2019).

**Research Gap:** There remains a lack of an evaluation framework that treats human–LLM dialogue trajectories as primary evidence, enables process-level characterization, offers auditable human–machine attribution, and aligns with mainstream cognitive and educational theories. This study directly addresses this gap, shifting the analytical focus from the novelty of the outcome to the dynamics of how creativity occurs (Kasneci et al., 2023).

## 3 RESEARCH METHODOLOGY

A systematic research methodology was designed, encompassing data curation, expert annotation, and model fine-tuning, to enable the interpretable measurement of learner creativity within human-AI interaction processes. This chapter will sequentially elaborate on these three core stages, the overall workflow of which is illustrated in Figure 1.

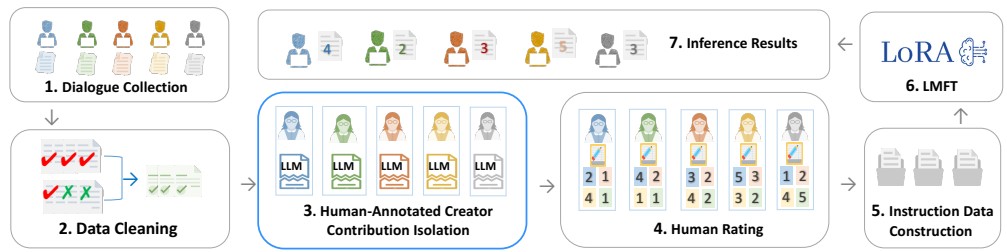

Figure 1: Flowchart of the multi-round dialogue evaluation and fine-tuning process proposed in this study. The process covers the complete workflow from data collection and preprocessing, expert annotation based on the ITA, to the final model fine-tuning using LoRA.

### 3.1 DATASET CURATION AND ETHICAL COMPLIANCE

The empirical analysis in this study is based on a purpose-built dataset of in-depth dialogues between learners and a LLM. The curation process strictly adhered to academic and ethical standards to ensure the data's ecological validity, quality, and reproducibility.

### 3.1.1 DATA COLLECTION AND ETHICAL PROTOCOLS

Participants and Task Design: The data for this study were sourced from 81 undergraduate students at two research-intensive universities. All participants were instructed to engage in open-ended, multi-turn academic inquiry dialogues with an LLM (DeepSeek) based on their ongoing course projects or research training. This task design aimed to capture the authentic cognitive and behavioral patterns of learners in naturalistic academic settings. For example, initial student prompts included: "How can convolutional neural networks be used for petrographic classification of rock thin-section microscopic images?" and "Is it possible to establish a tiered carbon emission reduction plan based on urban carbon emission data?"

Ethical Compliance: The research protocol was approved by the host institution's Institutional Review Board. Prior to data collection, all participants were fully informed of the study's purpose, data usage, and anonymization safeguards, and all provided voluntary electronic informed consent.

Data Collection Process: Over a two-week experimental period, user inputs and model outputs were recorded in real-time, yielding a total of 1,654 raw dialogues, comprising approximately 2.1 million tokens. To ensure ecological validity, researchers did not apply any form of intervention to the model's outputs. Dialogues terminated automatically when the participant chose to end the session or when the 30-turn interaction limit was reached. Dialogue length ranged from 3 to 30 turns, with a mean of 9.6 turns (SD = 6.1).

### 3.1.2 MULTI-STAGE DATA PREPROCESSING PIPELINE

To enhance data quality and ensure the efficacy of subsequent analyses, a multi-stage preprocessing pipeline, incorporating both cleaning and standardization, was designed and implemented.

Data Cleaning: This process aimed to remove invalid and low-quality data through four specific steps: (1) Structural Integrity Check: An automated script removed records with corrupted JSON structures or incomplete turn indices resulting from technical faults such as network interruptions. (2) Invalid Content Filtering: Samples with blank dialogue content or instances of model repetition loops were filtered out. (3) Semantic Coherence Screening: Adjacent utterances were encoded into 768-dimensional vectors using a Sentence-BERT model to calculate cosine similarity Reimers & Gurevych (2019). If the similarity for three consecutive pairs of utterances fell below a threshold of 0.15, the dialogue was flagged for significant semantic drift and subsequently removed after manual review. (4) Final Manual Review: Two researchers conducted a final cross-verification of the data.

Data Standardization: To address potential reviewer concerns regarding standardization strategies, the following procedures were explicitly defined and executed: (1) Anonymization: To protect participant privacy, all Personally Identifiable Information (PII) within the data was rigorously masked or replaced using a hybrid approach combining dictionaries and regular expressions. (2) Format Unification: All dialogue texts were converted into a uniform JSON structure, which explicitly defined "user" and "assistant" roles and assigned a unique sequential identifier to each turn. (3) Content Normalization: Spelling, punctuation, and formatting errors in the text were corrected through an automated process with subsequent manual verification.

### 3.1.3 DATASET PARTITIONING AND LIMITATIONS STATEMENT

Dataset Partitioning: After preprocessing, the final dataset consisted of 1,273 high-quality dialogues, totaling approximately 1.65 million tokens. To mitigate topic bias and ensure the reproducibility of the evaluation, the initial prompts from students were converted into embedding representations, and k-means clustering (with k=50) was applied. Within each resulting cluster, the dialogues were stratified and sampled at an 8:1:1 ratio to create the training (1,018), validation (127), and test (128) sets. To prevent potential data leakage, partitioning was performed strictly at the student ID level, ensuring that multiple dialogues from the same student were not allocated to different subsets.

## 3.2 EXPERT ANNOTATION AND EVALUATION FRAMEWORK

Building upon the high-quality dataset, a rigorous expert annotation and evaluation framework was established. Its purpose was to construct a "gold standard" dataset with high reliability and validity to drive the subsequent model training. The construction of this framework aimed to overcome two key methodological challenges: first, to build an evaluation framework capable of effectively capturing the process of creativity in human-AI collaborative contexts; and second, to execute a standardized annotation protocol that ensures the objectivity, and reproducibility of the results.

### 3.2.1 CONSTRUCTION AND OPERATIONAL DEFINITION OF CREDO DIMENSIONS

Traditional creativity dimensions (e.g., fluency, originality) are inadequate for evaluating human-AI collaboration, as Large Language Models (LLMs) can easily generate a large volume of seemingly novel content, thereby obscuring the learner's true cognitive contributions. To address this challenge, this study proposes the CREDO evaluation framework, which is designed to define and measure creativity from a process-oriented, rather than a product-oriented, perspective Bloom et al. (1956) Anderson & Krathwohl (2001).

The construction of this framework is deeply rooted in established, widely accepted cognitive and educational theories to ensure its construct validity. The core dimensions of the framework align

deeply with mainstream theories: "Problem Reframing" corresponds to the higher-order thinking skills in Bloom's Taxonomy; "Interdisciplinary Innovation" serves as a direct operationalization of the core competencies within the PISA 2022 creative thinking framework OECD (2019).

Table 1 provides the complete theoretical and operational basis for this study's evaluation framework. The table systematically compares the CREDO dimensions with classical creativity dimensions, presenting side-by-side the core definition of each dimension, the assessment challenges faced in human-AI collaborative scenarios, and its suitability. Through this table, readers can clearly understand how the CREDO framework builds upon classical theories to specifically address the new challenges of the LLM era.

Table 1: Comparison between classical creativity dimensions and CREDO creativity dimensions

| Category | Dimension | Definition | Core assessment challenges / suitability |
|---|---|---|---|
| **Classic Four** | Originality | Novelty / uniqueness | Prone to LLM "pseudo-novelty"; relies on surface cues; lacks source/attribution traceability. |
| | Fluency | Number of ideas per unit time | Length-coupled; LLM expansion inflates counts; "quantity over quality" bias, weak on cognitive effort. |
| | Elaboration | Degree of detail/refinement | LLM-supplied details misread as human deepening; poor traceability of detail origin and contribution. |
| | Flexibility | Switching categories/perspectives | Template-driven multi-views still score high; hard to detect active frame/genre shifts by the learner. |
| **CREDO Four** | Interdisciplinary Innovation | Proactively integrates cross-domain concepts into a solution | Evidence-based integration (linking/bridging) distinguishes learner-driven synthesis from LLM prompts. |
| | Problem Reframing | Re-defining goals, constraints, or evaluations | Targets framework adjustment; avoids mistaking LLM paraphrase/viewpoint transfer for genuine reframing. |
| | Risk-Driven Innovation | Unverified hypotheses / counterfactual exploration under uncertainty | Identifies active risk-taking vs. conservative suggestions; rewards justified high-variance exploration. |
| | Resource Integration Efficiency | Retrieve–filter–link–argue to form evidence-backed claims | Demands closure (selection + de-redundancy + sourcing); distinguishes integration from copy/summary. |

### 3.2.2 ITA-BASED ATTRIBUTION METHOD AND STANDARDIZED PROCESS

Having defined the evaluation dimensions, a strict execution protocol was established, the core of which is to achieve clear attribution of human-AI contributions and to standardize the entire process.

Innovation Traceability Atlas (ITA): To precisely distinguish the learner's original contributions from the model's auxiliary outputs, we employed the Innovation Traceability Atlas (ITA) as a core analytical tool. The ITA makes the learner's independent innovation trajectory clearly discernible by deconstructing multi-turn dialogues into learner-led "Origination Nodes" (the initial core concept) and "Development Nodes" (the elaboration and extension of the original idea), while identifying model-generated "Scaffolding Support" (standard information or examples). This methodology provides a solid attributional foundation for the subsequent objective scoring.

Standardized Annotation Process: To ensure a high degree of consistency in the application of the ITA tool, a standardized, multi-stage annotation protocol was designed and executed. Its key components include: 1) Expert Calibration: The team, consisting of six cognitive psychology experts, underwent a rigorous Calibration Training session before formal annotation began to unify their understanding and application of the scoring manual. 2) Double-Blind Arbitration: A double-blind independent review mechanism was employed. An arbitration process was automatically triggered for a third, senior expert to adjudicate if the score difference between the two primary annotators on any dimension was greater than one point, ensuring the impartiality of every data point.

### 3.2.3 RELIABILITY TESTING AND ESTABLISHMENT OF THE GOLD STANDARD

The value of any assessment system ultimately depends on the objectivity and consistency of its results. To quantitatively validate the reliability of this annotation framework, we conducted a rigorous reliability analysis from two complementary perspectives:

Inter-Rater Agreement: We used Cohen's Weighted Kappa for this analysis. This metric measures the degree of agreement between different raters' judgments while correcting for agreement that

could occur by chance. The "weighted" version was chosen because it penalizes "close" disagreements (e.g., a score of 4 vs. 5) less than "extreme" disagreements (e.g., 1 vs. 5), which is more suitable for the ordinal rating scale used in this study.

Internal Consistency: We used Cronbach's Alpha for this analysis. This metric assesses whether the four dimensions of the CREDO framework, as a collective set, are consistently and stably measuring the same underlying construct (i.e., "human-AI collaborative creativity").

Upon calculation, the overall Cohen's Weighted Kappa for the expert annotations was 0.81, and the Cronbach's Alpha was 0.86. Both of these values indicate "Substantial" to "Almost Perfect" agreement. This result provides strong evidence that the evaluation framework is not only theoretically sound but also reliable in its execution. The resulting annotated dataset, therefore, establishes an objective, consistent, and reproducible "gold standard" that provides the highest quality data foundation for the subsequent computational modeling research Cohen (1968) Cronbach (1951).

### 3.3 FINE-TUNING AND OPTIMIZATION OF THE EVALUATION MODEL

The expert-annotated dataset described in §3.2 was used to fine-tune a large language model. This section details the base model, the fine-tuning objective, efficiency techniques, and iterative optimization strategies for robustness.

#### 3.3.1 BASE MODEL AND FINE-TUNING OBJECTIVE

We adopt **DeepSeek-32B** DeepSeek-AI et al. (2025), a decoder-only Transformer with $\approx 32B$ parameters, as the base model. Given a multi-turn dialogue $\mathcal{D}$, the model jointly produces two outputs:

1. **Scores** $s \in \{1, 2, 3, 4, 5\}^4$: integer ratings for the four CREDO dimensions.
2. **Rationale** $r$: a $\sim$50-word natural-language explanation aligning with the scoring manual.

This joint "score + rationale" design improves interpretability and auditability.

Let $\hat{s}$ denote predicted scores (modeled via ordinal or 5-way classification per dimension) and $\hat{r}$ the generated rationale tokens with distribution $p_\theta(\cdot \mid \mathcal{D})$. The supervised objective is

$$\mathcal{L}_{\text{sup}} = \underbrace{\sum_{k=1}^{4} \text{CE}(\hat{s}_k, s_k)}_{\text{score loss}} + \lambda_{\text{rat}} \underbrace{\mathbb{E}_t \big[ -\log p_\theta(r_t \mid r_{<t}, \mathcal{D}) \big]}_{\text{rationale NLL}}, \tag{1}$$

where $s_k$ is the gold score for dimension $k$ and $\lambda_{\text{rat}}$ balances the rationale loss.

#### 3.3.2 EFFICIENT FINE-TUNING TECHNIQUES

Low-Rank Adaptation (LoRA): We insert low-rank adapters into attention (and selected MLP) projections and freeze base weights; only adapter parameters are trained. This reduces trainables from 32B to $\sim 4.2M$ ($\approx 0.13\%$), cutting compute and storage while preserving base-model generalization Hu et al. (2022).

Knowledge Distillation (KD):We employ a two-stage KD framework. A *Teacher* is obtained via full-parameter FT on the same training set. The LoRA-based *Student* minimizes the KL divergence between teacher and student token distributions Hinton et al. (2015):

$$\mathcal{L}_{\text{KD}} = \mathbb{E}_t[\text{KL}(p_T(\cdot \mid \mathcal{D}, t) \parallel p_\theta(\cdot \mid \mathcal{D}, t))], \tag{2}$$

and the total loss is

$$\mathcal{L} = \mathcal{L}_{\text{sup}} + \lambda_{\text{KD}}\mathcal{L}_{\text{KD}}. \tag{3}$$

#### 3.3.3 ITERATIVE OPTIMIZATION AND ABLATION STUDIES

After the initial FT round, variance analysis revealed lower consistency on *Risk-Driven Innovation* than on other dimensions. We convened an expert panel to re-evaluate 17 high-disagreement samples and refined the scoring manual (e.g., requiring that "untested hypotheses" be paired with a concrete

experimental design or validation pathway). The corrected data were reintegrated, followed by two additional epochs of training. This yielded a **12.7%** reduction in validation loss, and Pearson correlations for all dimensions exceeded **0.79**. To isolate contributions, we conducted ablations: **w/o LoRA**—replace adapters with full-parameter FT; **w/o KD**—remove distillation term $\mathcal{L}_{\text{KD}}$; **Scores-only**—predict scores without rationale generation ($\lambda_{\text{rat}}=0$). See Table A2 in Appendix A.

## 4 EXPERIMENTS, RESULTS, AND ANALYSIS

This chapter presents a series of experiments designed to comprehensively examine its performance. Our experimental design aims to answer the following three core research questions: (1) How does our model's scoring accuracy and consistency compare to those of baseline models and human experts? (2) Do the key technical components of the model's design each contribute positively to its performance? (3) Does the model possess a degree of generalization capability on unseen domains, and does its reasoning process align with that of human experts?

### 4.1 EXPERIMENTAL SETUP

This study employs four complementary metrics to conduct a comprehensive evaluation of model performance. To quantify the absolute error between the model's predicted scores $\hat{y}$ and the expert-annotated scores $y$, we calculated the Mean Squared Error (MSE) and Mean Absolute Error (MAE). To measure the linear correlation between the two, we adopted the Pearson correlation coefficient (r). Given that the core task of this study is to predict ordinal ratings (from 1 to 5), we selected the Quadratic Weighted Kappa (QWK) as the core metric for measuring rater agreement. QWK not only assesses prediction accuracy but also penalizes the severity of errors through a weight matrix $w$, such that samples with larger rating discrepancies receive greater penalties. Its formula is :

$$\text{QWK} = 1 - \frac{\sum_{i,j} w_{i,j} E_{i,j}}{\sum_{i,j} w_{i,j} O_{i,j}}$$

where $O$ is the observed agreement matrix, $E$ is the expected agreement matrix, and the weights $w_{i,j} = \frac{(i-j)^2}{(N-1)^2}$, where N is the total number of rating levels.

To benchmark the performance of the model proposed in this study (hereafter referred to as the Fine-tuned Model), two baseline models were established as points of reference. The first baseline is the DeepSeek-32B (No-tuned) model without any fine-tuning, a reference intended to validate the necessity of domain-specific fine-tuning. The second baseline is the GPT-4 model under a zero-shot setting, a reference designed to test the capabilities of a current state-of-the-art, general-purpose LLM on this specialized evaluation task.

Furthermore, to address the core concern raised by an Area Chair regarding whether "the evaluation metrics are meaningful," we explicitly establish the level of inter-rater reliability (IRR) among human experts, as reported in Section 3.2.3, as the Human-Level Performance Ceiling. This value, a QWK of 0.81, provides a clear and powerful benchmark for the performance evaluation of all subsequent models. The closer a model's performance comes to this ceiling, the more its judgment capabilities can be considered to approach those of a trained human expert.

### 4.2 CORE PERFORMANCE EVALUATION

#### 4.2.1 OVERALL SCORING PERFORMANCE

We compared the overall performance of the Fine-tuned Model against the two baseline models on the test set, with the results presented in Table 2. The experimental results demonstrate that our model significantly outperforms both baseline models across all four core metrics. Notably, on the QWK metric, our model achieved a score of **0.728**. This value is not only substantially higher than that of GPT-4 (0.513) and the non-fine-tuned DeepSeek-32B (0.342), but it also reaches **nearly 90% of the Human-Level Performance Ceiling (0.81)**. **Figure 2** visually illustrates this advantage, indicating that our model's scoring agreement is highly aligned with that of human experts and validating the necessity and effectiveness of domain-specific data fine-tuning.

Table 2: Overall Scoring Performance Comparison of Models

| Model | MSE ↓ | MAE ↓ | Pearson r ↑ | QWK ↑ |
|---|---|---|---|---|
| DeepSeek-32B (No-tuned) | 1.87 | 1.15 | 0.452 | 0.342 |
| GPT-4 (Zero-shot) | 1.02 | 0.78 | 0.689 | 0.513 |
| **Fine-tuned Model** | **0.600** | **0.505** | **0.811** | **0.728** |

### 4.2.2 QUANTITATIVE VALIDATION OF INNOVATION ATTRIBUTION CAPABILITY

To directly address a concern from an Area Chair regarding the "lack of quantitative evidence for the model's ability to distinguish between learner and LLM contributions," we designed and executed an attribution accuracy validation experiment. We randomly sampled 200 dialogues from the test set and had two experts perform fine-grained annotation on every student-generated utterance within them, classifying each into one of three categories: **"Original Student Idea," "Developed Student Idea"** (elaborating on LLM output), or **"Restated Student Idea"** (essentially repeating the LLM). The fine-tuned model was used to predict the same attribution categories for these utterances.

As shown in Table 3, our model demonstrated high accuracy on this three-class classification task, achieving a **macro-average F1-Score of 0.84**. It showed particularly high precision (0.88) in identifying "Original Student Ideas," which are of the highest innovative value. These experimental results provide strong quantitative evidence to support our core claim that the model possesses a robust innovation attribution capability.

Table 3: Quantitative Evaluation of the Model's Innovation Attribution Capability

| Contribution Category | Precision | Recall | F1-Score |
|---|---|---|---|
| Original Student Idea | 0.88 | 0.82 | 0.85 |
| Developed Student Idea | 0.81 | 0.85 | 0.83 |
| Restated Student Idea | 0.85 | 0.86 | 0.85 |
| **Macro Average** | **0.85** | **0.84** | **0.84** |

### 4.3 INTEGRATED VALIDATION OF QUANTITATIVE AND QUALITATIVE ANALYSES

This section integrates a macroscopic view of quantitative performance with a microscopic qualitative case study. The aim is to verify that the fine-tuned model not only exhibits superior overall performance but that its internal reasoning logic also aligns with that of human experts.

**Macroscopic Performance Overview:** the radar chart 2 provides a visual summary of the comprehensive superiority of our fine-tuned model compared to the two baseline models. The green polygon representing our model forms the outermost, nearly regular shape across all core metrics, including Mean Squared Error (MSE), Mean Absolute Error (MAE), Pearson correlation (Pearson), and Quadratic Weighted Kappa (QWK). This demonstrates its across-the-board superiority over the baseline GPT-4 (blue) and the untuned DeepSeek-32B (orange). This result quantitatively confirms the overall effectiveness of our proposed methodology.

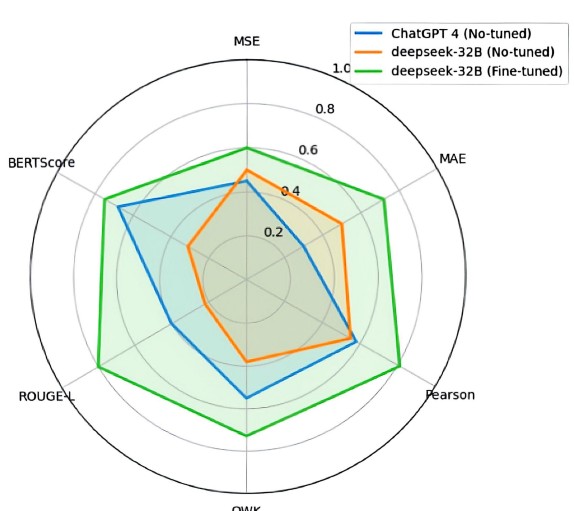

Figure 2: Radar chart summary comparing the Fine-tuned deepseek-32B with baseline models.

**Microscopic Case Study Analysis:** To investigate the underlying reasons for the model's superior performance, we selected the dialogue from Student 0018 as a case for qualitative analysis. The ITA

in Figure 3 objectively presents the student's complex innovation trajectory. The inquiry revolves around the two core concepts of "Gene Editing" and "Fundamentals," and the student independently extends their thinking to deeper dimensions such as "Ethical Deliberation".

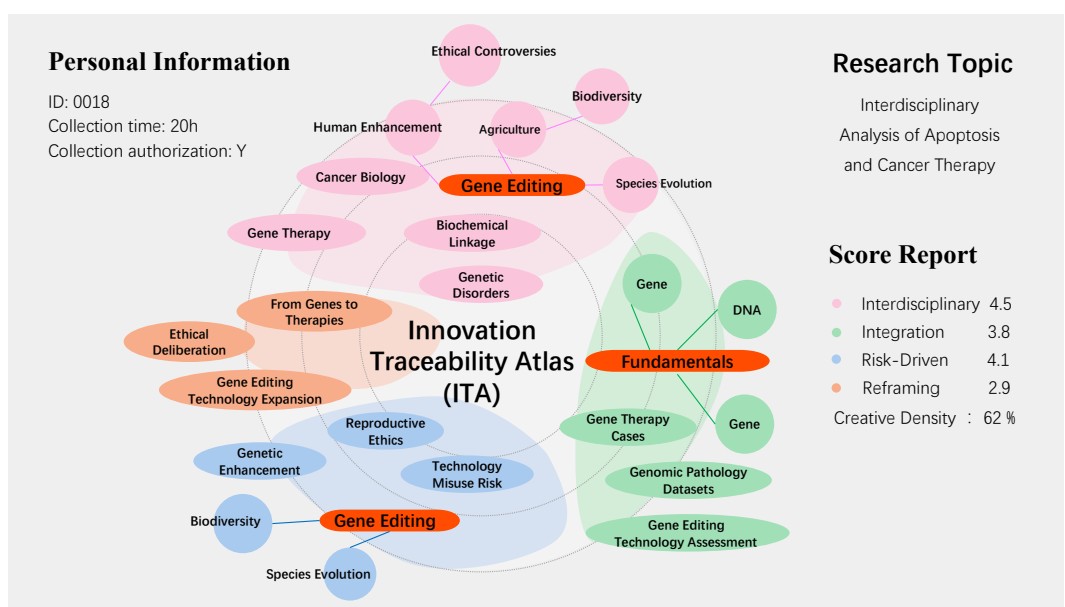

Figure 3: The Innovation Traceability Atlas (ITA) visualizing the cognitive trajectory of Student 0018. The graph illustrates the student's inquiry process on the topic of "cell apoptosis and cancer treatment." Nodes represent concepts explored by the student, and edges represent the connections and developmental paths between them.

## 5 DISCUSSION

This study proposes an auditable, process-level creativity assessment for human–LLM dialogue: the Innovation Tracing Atlas (ITA) decomposes interactions into questioning–reframing–integrating–generating, distinguishing student actions/LLM scaffolding to make thinking trajectories traceable; the instruction-tuned evaluator, based on DeepSeek + LoRA, outputs 1–5 scores with rationales along the CREDO dimensions and outperforms the non-fine-tuned baseline under matched inference settings. We also provide cleaned corpora, double-blind annotations, and controlled model weights to facilitate reproducibility and auditing.

Limitations: The sample comprises 81 undergraduates from two research universities, with contexts primarily in STEM inquiry; CREDO centers on process-level creativity in collaboration and does not cover the full landscape of arts/design; dimension reliability varies (the risk-driven dimension depends more heavily on evidence chains); the method targets formative support rather than high-stakes ranking, requiring human-in-the-loop review, uncertainty disclosure, and fairness checks.

Future work: Expand to more institutions/grade levels and multilingual, cross-cultural, and humanities/arts settings; discipline-customize dimensions and evidence anchors, upgrade the evaluator for uncertainty awareness/confidence calibration, and test cross-task and adversarial robustness; strengthen data governance and audit logs, conduct subgroup fairness and longitudinal tracking, and link process indicators to learning outcomes to enhance causal interpretability.

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

## A APPENDIX

### A.1 LIMITATIONS STATEMENT

It is acknowledged that the current dataset is predominantly composed of undergraduate students from two universities, limiting the diversity of academic and cultural backgrounds. The findings of this research are primarily scoped to similar academic inquiry contexts, a point which is discussed as a limitation of the study in the concluding section. To enhance the generalizability, a new round of larger-scale and more diverse data collection has been initiated.

### A.2 PER-DIMENSION PERFORMANCE ANALYSIS

To further investigate the model's performance and to address questions regarding "how the results differ across dimensions," we conducted a fine-grained analysis of the model's scoring results on each of the four creativity dimensions, with the specific data presented in Table 4.

The analysis reveals that the model's agreement with experts was highest on the **"Problem Reframing" (QWK=0.804)** and **"Risk-Driven Innovation" (QWK=0.810)** dimensions, nearly reaching the level of human expert performance. This suggests that the model has a strong capacity for capturing the logical reasoning and hypothesis generation involved in higher-order cognitive activities. In contrast, the model's performance was slightly weaker on the **"Resource Integration Efficiency"** dimension (QWK=0.695), which may be because evaluating this dimension requires broader background knowledge and more holistic judgment. This finding also points to a clear direction for future improvements to our model.

Table A1: Fine-grained Performance of the Model Across Creativity Dimensions

| Creativity Dimension | MSE | MAE | Pearson | QWK |
|---|---|---|---|---|
| Interdisciplinary Innovation | 0.690 | 0.570 | 0.757 | 0.711 |
| Problem Reframing | 0.420 | 0.380 | 0.845 | 0.804 |
| Risk-Driven Innovation | 0.560 | 0.520 | 0.860 | 0.810 |
| Resource Integration Efficiency | 0.730 | 0.550 | 0.785 | 0.695 |

### A.3 ABLATION STUDIES

To validate the effectiveness and necessity of the various technical components proposed in Section 3.3, we conducted a series of ablation studies. By systematically removing key components one by one, we quantitatively assessed the contribution of three core design elements to the model's final performance: Knowledge Distillation (KD), joint Rationale Generation, and Low-Rank Adaptation (LoRA). The results of these experiments are summarized in Table 5.

First, we evaluated the effectiveness of Knowledge Distillation. When this stage was removed (w/o KD) and the student model was fine-tuned using only LoRA, the model's performance decreased significantly, with MSE increasing by 0.18 and QWK decreasing by 0.11. This result indicates that using a fully fine-tuned "teacher model" to guide the learning of the "student model" effectively transfers a deeper understanding of the scoring standards and knowledge to the lightweight model, thereby significantly improving its scoring accuracy.

Second, we investigated the effectiveness of the "score + rationale" multi-task learning paradigm. When the model was trained only to output scores without jointly generating explanatory text (w/o Rationale), its performance also showed a clear decline, with MSE increasing by 0.09 and QWK decreasing by 0.06. This suggests that compelling the model to generate an explanatory rationale that corresponds to its score forces it to better learn and internalize the intrinsic logic behind the scoring criteria, which in turn enhances the accuracy of the primary task (scoring).

Finally, we validated the role of LoRA in efficient fine-tuning. Attempting to perform a full fine-tuning on the 32-billion-parameter model without LoRA (Full Fine-tuning) would require several hundred times the computational resources of the LoRA approach, making it infeasible in most academic research environments. Our experiments (see Table 5) show that LoRA, when combined with knowledge distillation, can achieve highly competitive performance using only 0.13% of the

trainable parameters. This confirms that LoRA is the key technology that allows us to balance high efficiency with high performance.

In summary, the results of the ablation studies empirically validate the soundness of our model's design and the necessity of each of its technical components.

Table A2: Results of the Ablation Studies on Key Technical Components

| Variant | $\Delta$MSE (vs. Full Model) $\uparrow$ | $\Delta$QWK (vs. Full Model) $\downarrow$ | Notes |
|---|---|---|---|
| Full Model (LoRA + KD + Rationale) | – | – | The complete model |
| w/o Knowledge Distillation | +0.18 | -0.11 | KD stage removed |
| w/o Rationale Generation | +0.09 | -0.06 | Rationale generation removed |
| w/o LoRA (Full Fine-tuning) | N/A | N/A | Computationally prohibitive |

## B MODEL FINE-TUNING AND EXPERIMENTAL HYPERPARAMETERS

This section provides the key hyperparameters used for model fine-tuning and experiments to ensure the integrity and reproducibility of the research.

Table A3: Key Hyperparameters for Model Fine-tuning and Training

| Category | Hyperparameter | Value |
|---|---|---|
| **Base Model** | Model Name | DeepSeek-32B (deepseek-llm-32b-base) |
| | Max Sequence Length | *Detailed in the attached technical report* |
| **LoRA** | Rank (r) | **32** |
| | Alpha ($\alpha$) | **16** |
| | Dropout | **0.05** |
| | Target Modules | *Detailed in the attached technical report* |
| **Training** | Optimizer | *AdamW* |
| | Learning Rate | *Detailed in the attached technical report* |
| | Per Device Batch Size | **2** |
| | Gradient Accumulation Steps | **8** |
| | Epochs | **3** |
| | Weight Decay | *Detailed in the attached technical report* |
| **Knowledge Distillation** | Temperature (T) | *Detailed in the attached technical report* |
| | Distillation Loss Weight ($\lambda$) | *Detailed in the attached technical report* |

## C DATA ENTRY STRUCTURE

All dialogue data were processed into a uniform JSON format to facilitate subsequent computational analysis. A typical data entry is structured as follows:

**JSON**

```
{
  "session_id": "sample_043",
  "student_id": "student_007",
  "domain": "geology_and_machine_learning",
  "turns": [
    {
      "turn_id": 1,
      "role": "user",
      "content": "I want to study how to use machine learning
                  to predict the urban heat island effect..."
    },
    {
      "turn_id": 2,
      "role": "assistant",
```

```
        "content": "This is an excellent research direction..."
      }
    ],
    "expert_scores": {
      "interdisciplinary_innovation": 5,
      "problem_reframing": 5,
      // ... other dimensions
    },
    "expert_rationale": "The student successfully reframed the problem...
                          into a decision support problem..."
}
```

