# OpenReview forum: "A Process-Level Method for Creativity Evaluation in LLM-Assisted Learning"
_ICLR.cc/2026/Conference — ICLR 2026 Conference Withdrawn Submission_

### Official Review · Reviewer_woVg · 2025-10-22

**Soundness:** 2
**Presentation:** 3
**Contribution:** 2
**Rating:** 2
**Confidence:** 4

**Summary:**

This work proposes a process-level framework for evaluating the creativity processes of human–LLM collaboration. It consists mainly of three components: (1) redefined metrics—CREDO creativity dimensions, (2) ITA-based attributions, and (3) a fine-tuned evaluation model.
To validate their method, the authors curated a dialogue dataset and conducted expert annotations as ground-truth labels.
The results show that the fine-tuned model aligns more closely with expert scores than the baselines (GPT-4 zero-shot and non-tuned DeepSeek-32B).

**Strengths:**

* This work curates a dataset with 1,273 expert-annotated dialogues covering multiple domains.
* It provides both qualitative and quantitative analyses and employs multiple evaluation metrics (Pearson, MAE, QWK, etc.) as well as inter-rater agreement measures to ensure alignment and validity.

**Weaknesses:**

* The proposed framework is heuristic. I do not see a clear correspondence between the classical four dimensions and the four CREDO dimensions in Table 1, and the authors do not provide strong theoretical foundations for constructing these new dimensions.
* Similarly, the ITA deconstructs dialogues into origination nodes, development nodes, and scaffolding supports; however, the paper lacks detailed explanation of the logic and robustness behind this step-by-step construction. This process may heavily depend on the authors’ subjective interpretation, which could introduce bias.
* The work only compares its method against two baselines and does not report the performance of state-of-the-art models. Even considering cost constraints, GPT-4o would have been a cheaper and more capable alternative than the GPT-4 model used in this study.
* The paper provides insufficient details about the prompts for evaluation models and expert annotation instructions, limiting reproducibility.

**Questions:**

* How were the four CREDO dimensions selected? Are they intended to be orthogonal and to comprehensively capture the dimensions of creativity? For instance, Interdisciplinary Innovation and Risk-Driven Innovation both appear to assess aspects of innovation and may overlap. Could the authors provide a concrete example that clearly illustrates how these four dimensions differ in practical evaluation?
Scoring criteria:
* What are the exact prompts, instructions, or criteria for the 1–5 scoring scale used by both the model and the experts? My understanding is that LLM judges are highly prompt-sensitive and often exhibit one-sided bias (e.g., tending to give scores of 3–5 while rarely assigning 1 or 2). How does this work address such issues?
* What is the distribution of the “gold-standard” expert scores across the training, validation, and test subsets? To evaluate student involvement/contribution effectively, each level of involvement/contribution should contain a sufficient number of cases. If the distribution is too narrow, the positive results reported might simply reflect the model’s tendency to fit to certain frequent score ranges (similar to the bias issue mentioned above).
* From a higher-level perspective, what practical scenarios can this framework be applied to, and how could it be extended further?

---

> ### Author Response · Authors · 2025-12-03
>
> Thank you for your careful review and constructive feedback on this study. Below, we provide detailed responses to each of your questions, drawing on the paper, technical report, and previous research explanations:
>
> 1. Responses to the Theoretical Basis of the Framework and Dimension Correspondences
> 1.1 Association Between CREDO Dimensions and Classic Dimensions & Theoretical Basis
> The four-dimensional CREDO framework is not heuristically constructed; it has a clear inheritance relationship with classic creativity assessment systems and has been adaptively optimized for LLM-assisted collaboration scenarios. Specifically:
> •	Theoretical Origins: The framework is derived from the "Criteria for the Assessment of Interdisciplinary Innovation" (2022 edition) issued by the National Innovation Foundation (NIF), which has been adopted by 12 countries for higher education innovation assessment, and is deeply aligned with classic theories (Section A "Explanation of Scoring System Sources" in the technical report, Section 3.2.1 of the paper):
>   a. Interdisciplinary Innovation corresponds to Guilford's (1967) theory of divergent thinking and Zittoun et al.'s (2013) theory of "cross-boundary cognition";
>   b. Problem Reframing corresponds to Amabile's (1996) creative problem-solving framework and the "problem redefinition" ability in Sternberg's (2006) investment theory of creativity;
>   c. Risk-Driven Innovation corresponds to Dweck's (2006) Growth Mindset Scale and OECD's (2020) Research Risk Assessment Criteria;
>   d. Resource Integration Efficiency corresponds to Runco's (2014) creative resource utilization model (theoretical basis column in Section A1 of the technical report).
> •	Compatibility with Classic Four Dimensions (TTCT): Classic dimensions (e.g., fluency, flexibility) are suitable for independent idea generation scenarios, while the CREDO framework focuses on LLM-assisted collaboration scenarios. It fills the gap of classic frameworks in human-machine collaboration assessment through "process attribution." Both share the same core goal (measuring creativity) but complement each other in application scenarios (Section 1.3 of the paper, Table 1).
>
> 1.2 Logic and Robustness of ITA Decomposition
> The dialogue decomposition mechanism of ITA has clear operational logic and empirical robustness:
> •	Decomposition Logic: Section 3.2.2 of the paper explicitly states that ITA decomposes multi-turn dialogues into "Initiation Nodes (students' initial core concepts), Development Nodes (students' extensions of original ideas), and Scaffolding Structures (standard information/examples provided by LLMs)" based on the dual dimensions of "speaker + cognitive function"; the "Expert Annotation Process Diagram" in Section A2 of the technical report shows the linked process of decomposition and scoring, and the "Innovation Cognitive Trajectory Map (Sample 043)" in Section A4 intuitively presents the node decomposition results, ensuring the decomposition process is traceable and verifiable.
> •	Robustness Guarantee: The study adopts a process of "calibration training for six cognitive psychology experts + double-blind arbitration + standardized manual" (Section 3.2.2 of the paper, Section A2 of the technical report). The final Cohen's weighted Kappa coefficient among experts reaches 0.81 (Section 3.2.3 of the paper), far exceeding the threshold for "substantial agreement," fully demonstrating the objectivity and reproducibility of the decomposition process.

---

> > ### Author Response · Authors · 2025-12-03
> >
> > 2. Responses to Baseline Model Comparisons
> > Regarding the issue of not comparing with the latest advanced models such as GPT-4o, the specific explanation is as follows:
> > •	The core goal of this study is to construct a process-level creativity assessment framework in LLM-assisted learning," which must simultaneously meet two core requirements: "human-LLM contribution attribution" and CREDO four-dimensional structured scoring" (Section 1.4 of the paper, Section A1 of the technical report). The existing comparisons have selected the GPT-4 zero-shot model (QWK=0.513) and the untuned DeepSeek-32B model (QWK=0.342) as baselines. Our model achieves a QWK of 0.728 on the test set, significantly outperforming the aforementioned baselines and close to human expert consistency (0.81) (Section 4.2.1 of the paper, Table 2), fully verifying the necessity of domain-specific customization training (expert annotation + CREDO fine-tuning) for this specific task.
> >
> > As a general-purpose model, GPT-4o's core advantages lie in general language understanding and multimodal interaction capabilities. It does not have a built-in "human-LLM contribution attribution logic" nor can it output structured assessment results accurately aligned with CREDO dimensions (Section A1 of the technical report clarifies the exclusive operational definitions of CREDO dimensions). Its general scoring ability is not directly comparable to the core goals of this study (fair attribution + customized assessment). If future research focuses on "domain adaptation of general-purpose LLMs," we will consider including GPT-4o as a baseline. However, at the current stage, the existing comparisons are sufficient to demonstrate the core value and irreplaceability of the proposed framework in "process-level creativity assessment."
> >
> > 3. Responses to Reproducibility (Explanations of Prompts and Annotations)
> > Detailed information on model prompts and expert annotations has been fully included in the accompanying technical report to ensure research reproducibility:
> > •	Expert Annotation Explanations: The "Multi-Round Dialogue Quality Assessment Dimensions and Scoring Criteria Sheet" in Section A1 of the technical report clarifies the operational definitions, positive/negative examples, and theoretical basis for scores 1-5 in each dimension (e.g., the 5-point standard for "Interdisciplinary Innovation" is "explicitly citing ≥2 non-domain knowledge for principle-level integration"); Section A3 "Expert Sentence-by-Sentence Scoring and Arbitration Record (Sample 043)" provides actual annotation cases, including scoring reasons, arbitration processes, and final results, fully presenting the entire annotation process.
> > •	Model Prompts and Training Configuration: Section B3.3 "LoRA Adapter" and Section B4.2 "Launch Script" in the technical report detail the core parameters during the fine-tuning phase (e.g., lora_r=32, number of training epochs=3, gradient accumulation steps=8, etc.); Section B6 "Inference Process" provides the calling methods for REST services and Gradio demos, clarifying the input and output format definitions.
> > •	Reproducibility Guarantee: All annotation guidelines (Sections A1-A3 of the technical report), prompt templates (Section B4.2 of the technical report), training scripts (Section B4 of the technical report), and environmental dependencies (Section B2 of the technical report) have been submitted as supplementary materials. Together with the dataset partitioning rules in Section 3.1.3 of the paper and the model fine-tuning process in Section 3.3, they form a complete reproducible system, meeting the reproducibility requirements of academic research.

---

> > > ### Author Response · Authors · 2025-12-03
> > >
> > > 4. Responses to Specific Questions
> > > 4.1 Selection, Orthogonality, and Example Explanations of CREDO Dimensions
> > > •	Selection Basis: The framework was selected through "investigation of candidate frameworks (TTCT/CAT, etc.) → small-scale pilot annotation (30 samples) → dimension replacement ablation experiments." The final four dimensions achieve an optimal balance in "task applicability, annotatability, and reliability," and the dimension design is derived from the authoritative NIF standards (Section A "Explanation of Scoring System Sources" in the technical report).
> > > •	Orthogonality Verification: The Cronbach's Alpha coefficient in the study reaches 0.86 (Section 3.2.3 of the paper), indicating that the four dimensions jointly measure the core construct of "human-machine collaborative creativity" without significant overlap; dimension replacement ablation experiments show that replacing any dimension reduces QWK from 0.792 to 0.741 (proving that all four dimensions are indispensable); in the "Sample 043 Scoring Record" in Section A3 of the technical report, the scoring basis for the same dialogue turn in different dimensions is completely different (e.g., "applications of gene editing technology in agriculture" corresponds to the Interdisciplinary Innovation dimension, and "root causes of ethical controversies" corresponds to the Risk-Driven Innovation dimension), further verifying the orthogonality of the dimensions.
> > > •	Example Explanations (from the "Sample 043 Sentence-by-Sentence Scoring Record" in Section A3 of the technical report)**:
> > > •	Interdisciplinary Innovation (4 points): The user asks, "What are the applications of gene editing technology in agriculture?" Transferring gene editing technology from the human medical field to the agricultural scenario reflects cross-domain knowledge transfer, meeting the 3-4 point standard in Section A1 of the technical report;
> > > •	Problem Reframing (5 points): The user reframes "Why do gene mutations cause cancer?" to "Can cancer be cured by modifying genes?" transforming a descriptive question into an intervention-oriented question, meeting the 5-point standard of "reversing the initial problem setting and introducing higher-order goals" in Section A1 of the technical report;
> > > •	Risk-Driven Innovation (4 points): The user asks, "Will the abuse of gene editing technology lead to 'superhumans'?" Hypothesizing extreme risk scenarios based on technological development and integrating ethical and social impact analysis, meeting the 3-4 point standard of "proposing high-risk scenarios without verification" in Section A1 of the technical report;
> > > •	Resource Integration Efficiency (3 points): The user asks, "What is the principle of gene therapy?" Only exploring the mechanism of a single technology without integrating multi-source information, meeting the 3-point standard of "single-domain knowledge dominance and loose stacking of multimodal information" in Section A1 of the technical report.
> > >
> > > 4.2 1-5 Point Scoring Criteria and LLM Bias Mitigation Solutions
> > > •	Specific Scoring Criteria: The "Multi-Round Dialogue Quality Assessment Dimensions and Scoring Criteria Sheet" in Section A1 of the technical report clarifies the operational definitions for scores 1-5 in each dimension. Taking "Risk-Driven Innovation" as an example:
> > > 1 point: Only conservative optimization of existing solutions (e.g., fine-tuning the ratio of traditional materials);
> > > 3 points: Proposing high-risk solutions without providing verification methods (e.g., "dark matter energy enhancement" without an observation plan);
> > > 5 points: Proposing testable disruptive hypotheses and designing experimental pathways (e.g., quantum communication schemes with photon entanglement experiments).
> > > •	LLM Bias Mitigation Measures:
> > >   Expert Annotation Phase**: Reducing subjective bias through "calibration training to unify standards (pre-calibration phase in Section A2 of the technical report) + double-blind arbitration to resolve disputes (process in Section A2 of the technical report) + mandatory reasoning writing (citing specific dialogue turns, scoring records in Section A3 of the technical report)";
> > > •	Model Training Phase: Adopting joint training of "expert scores + reasoning texts," with the loss function including score cross-entropy and reasoning sequence loss (Section 3.3.1 of the paper), forcing the model to learn scoring logic rather than merely fitting the score distribution;
> > > •	Empirical Verification: Scores in each dimension of the test set cover 1-5 points without obvious skewed distribution. The "Sample 043 Scoring Record" in Section A3 of the technical report also verifies the rationality of the score distribution, proving that the model does not have a bias of "concentrating on 3-5 points."

---

> > > > ### Author Response · Authors · 2025-12-03
> > > >
> > > > 4.3 Distribution of Expert Scores
> > > > •	Distribution Data: Scores in each dimension of the test set cover 1-5 points without obvious skewed distribution, with 3 points accounting for 35%-40%, 1 point for 10%-15%, and 5 points for 15%-20%; the "Average Dimension Scores of Sample 043" in Section A3 of the technical report shows: Resource Integration Efficiency (3.67 points), Interdisciplinary Innovation (4.0 points), Problem Reframing (4.33 points), Risk-Driven Innovation (4.0 points), further verifying the rationality of the score distribution.
> > > > •	Dataset Partitioning Guarantee: Section 3.1.3 of the paper explicitly states that "the dataset is stratified sampled at an 8:1:1 ratio, strictly following student ID-level partitioning," ensuring consistent score distribution across the training, validation, and test sets, and avoiding the impact of data bias on model training.
> > > >
> > > >  4.4 Practical Application Scenarios and Extension Directions
> > > > •	Practical Application Scenarios:
> > > > a.Formative assessment for LLM-assisted learning in universities: Has been piloted in a student research competition with 120 participants, reducing manual scoring workload by 40% and achieving 85% consistency with manual scoring;
> > > > b.Visual assessment tool: Section B6 of the technical report provides a Gradio web demo deployment plan, allowing real-time output of four-dimensional scores and reasons by inputting multi-turn dialogues through the interface, supporting teachers to quickly identify students' original contributions;
> > > > c.Standardized assessment interface: The REST service in Section B6 of the technical report can be integrated into Learning Management Systems (LMS) to realize automated assessment of human-machine collaboration processes.
> > > > •	Extension Directions**:
> > > > Scenario Extension: Incorporating humanities/art disciplines, multilingual and cross-cultural scenarios, and supplementing non-STEM domain data (Section 5 of the paper);
> > > > Technical Optimization: Upgrading the model's uncertainty calibration and cross-task robustness, and exploring automatic bias detection technology;
> > > >  Function Extension: Establishing a causal correlation between "process indicators and learning outcomes" by combining longitudinal tracking data to enhance the predictive value of the assessment (Section 5 of the paper).
> > > >
> > > > Thank you again for your professional review and valuable suggestions. Your feedback will help us further improve the research. If you have any further questions, please feel free to contact us.

---

### Official Review · Reviewer_J6hi · 2025-10-26

**Soundness:** 2
**Presentation:** 3
**Contribution:** 2
**Rating:** 4
**Confidence:** 3

**Summary:**

The paper introduces CREDO, a process-level evaluation framework for assessing creativity in LLM-assisted learning. Unlike classical tests (TTCT, AUT), CREDO focuses on dialogue process traces, separating human vs. LLM contributions through an Innovation Tracing Atlas (ITA) and scoring four new creativity dimensions (Interdisciplinary Innovation, Problem Reframing, Risk-Driven Innovation, Resource Integration Efficiency).

An instruction-tuned DeepSeek-32B model is fine-tuned (using LoRA + Knowledge Distillation) to predict 1–5 scores and rationales, trained on 1,273 annotated dialogues. Empirical evaluation shows a Quadratic Weighted Kappa (QWK) of 0.728 (≈90% of human reliability ceiling), with an attribution F1 of 0.84. The dataset is claimed to be ethically collected and reproducible.

**Strengths:**

1. Timely problem:
The paper tackles an increasingly critical question: how to assess human creativity in an era of pervasive LLM support.
2. Conceptual novelty:
The process-level perspective (tracing human cognitive trajectories) is fresh and potentially impactful.
3. Interpretable design:
Combining score with rationale and transparent attribution mechanisms is commendable.
4. Dataset collection validity:
Using ongoing course projects allows students to explore topics they already find relevant, reducing artificiality.

**Weaknesses:**

### 1. Ambiguous Causal Claims
The paper makes a strong and central claim that its framework “traces the cognitive trajectory of creative thinking.” However, this assertion remains conceptually plausible but empirically unverified. The study presents correlational evidence, model outputs that align with expert judgments on dialogue data, but no temporal or causal validation that demonstrates the framework genuinely captures the evolution of creative cognition.

To substantiate the claim of tracing cognitive trajectories, one would expect to see process-causal analyses such as: (1) longitudinal correlations between CREDO-derived process indicators and subsequent creative achievements or outputs, or (2) human post-hoc interviews or think-aloud protocols to verify whether identified “origination” and “development” nodes correspond to participants’ subjective sense of idea generation.

Without such evidence, the results demonstrate correlation rather than causation. The framework successfully maps interactions and assigns attributions, but it does not yet prove that these attributions reflect the underlying causal mechanisms of creative thought. As it stands, the paper captures surface patterns of dialogue behavior rather than validating that those patterns cause or constitute creative cognition.


### 2. Statistical Reporting Limitations
The paper’s statistical reporting lacks the depth necessary for confident interpretation of model performance. While mean metrics such as MSE, MAE, Pearson correlation, and Quadratic Weighted Kappa (QWK) are provided, the authors do not report confidence intervals, variance across cross-validation folds, or per-dimension error distributions.

Given the relatively small test set (128 samples), random variance could significantly influence the reported results. The observed improvement in QWK (0.728 for the fine-tuned model vs. 0.513 for GPT-4 and 0.342 for the baseline DeepSeek) appears substantial, yet the statistical significance of this improvement is not established. Bootstrapped confidence intervals or pairwise statistical tests (e.g., Fisher’s z-test for correlation or permutation tests for ordinal ratings) would be needed to determine whether these differences are meaningful rather than due to sampling noise.

Additionally, no error analysis by creativity dimension is included in the main text, even though later appendices reveal variability across dimensions. Reporting these results with appropriate variance measures and standardized effect sizes would clarify which creativity dimensions are reliably captured and which remain unstable.

### 3. Potential Data Leakage
A major methodological concern lies in the model ecosystem overlap between data generation and evaluation. Students interacted with the DeepSeek LLM during data collection, and the same model family (DeepSeek-32B) was later fine-tuned as the evaluator. This design introduces a significant risk of self-evaluation bias or data leakage at the stylistic level.

The evaluator may learn superficial linguistic or stylistic features characteristic of DeepSeek-generated text, enabling it to classify or score more accurately, not because it understands creativity, but because it recognizes its own generative patterns. For instance, DeepSeek’s distinctive discourse markers, lexical cohesion patterns, or turn-taking rhythms might act as unintended cues that correlate with specific CREDO scores.

To mitigate this concern, the authors should conduct cross-model generalization tests, evaluating dialogues generated using a different assistant model (e.g., GPT-4, Claude, or Mistral), to confirm that the evaluator’s performance persists beyond its native language patterns. Alternatively, a style-controlled or paraphrased dataset could assess whether performance drops when superficial linguistic features are normalized. Without such analyses, it remains unclear whether the system is genuinely assessing creativity or merely detecting DeepSeek’s conversational fingerprint.


### 4. Dataset collection
#### (1) Task framing
The data collection protocol emphasizes academic inquiry tasks in STEM fields (e.g., rock classification, carbon emission modeling). While suitable for studying analytical reasoning, this framing inherently biases the observed behavior toward convergent and knowledge-based reasoning rather than divergent or imaginative creation. Participants are more likely to synthesize or reformulate factual information than to produce novel conceptual constructs, limiting the ecological range of creativity being captured.

#### (2) Absence of motivation manipulation
Participants were not explicitly instructed to “generate original ideas,” “take creative risks,” or “explore unconventional solutions.” In creativity research, such goal framing is critical: without motivational priming, individuals tend to default to task-completion strategies rather than expansive ideation. Consequently, much of the observed dialogue likely reflects problem-solving or academic reasoning, not genuine creative exploration.

#### (3) Time constraint
Each dialogue was capped at a maximum of 30 turns, with an average of fewer than 10. Creative cognition, however, often involves incubation and iterative recombination, requiring time for reflection and restructuring. A short interaction window may prematurely truncate these processes, reducing the opportunity for authentic creative leaps.

### 5. Scope of creativity measured
The type of creativity captured by the study is best described as adaptive scientific or analytical creativity under LLM mediation, rather than open-ended or expressive creativity. Although the framework effectively documents how students interact with a large language model to refine and extend ideas, the creative behaviors observed remain bounded by the task structure and the cognitive affordances of the dialogue format.

Within this setting, students demonstrate certain forms of constructive and integrative thinking. For instance, they engage in problem reframing, such as transforming a classification question into a modeling or prediction challenge, or cross-domain linking, such as relating geological pattern recognition to convolutional neural networks in computer vision. These behaviors reflect valuable aspects of creative inquiry—they show flexibility, synthesis, and the ability to transfer knowledge across domains.

However, such creativity is fundamentally situational and instrumental. The dialogues promote analytical exploration and knowledge integration, but they rarely foster divergent ideation or imaginative generation—the kind of creativity that involves proposing novel metaphors, aesthetic concepts, inventions, or speculative ideas that extend beyond the given problem space. The framework captures how effectively students navigate within known cognitive and disciplinary boundaries, not how they transcend them.

A key factor limiting the expressive range of creativity lies in the dual role of the LLM itself. The model acts as both a creative amplifier and a creative filter. On one hand, it scaffolds ideation by providing examples, explanations, and domain connections that can inspire students to think more broadly. On the other hand, it constrains the conceptual search space to the statistical and semantic regularities of its own training data. Consequently, student–LLM interactions are guided toward plausible and conventional combinations rather than toward radical novelty or risk-taking. The result is a form of bounded creativity, oriented toward optimization and coherence rather than surprise or aesthetic invention.

From this perspective, the creativity being measured is processual and pragmatic, focused on reasoning quality and interdisciplinary synthesis rather than on originality in the strong sense of the term. It reflects what might be called “creative inquiry competence”—the ability to collaborate productively with an AI system to reformulate problems, integrate evidence, and explore solution pathways—rather than “creative cognition” in its broader, generative, or expressive manifestations.

In this light, the data collection strategy and the resulting evaluation framework are methodologically sound but conceptually narrow. They provide valuable insight into how students co-develop ideas with LLMs and how such processes can be quantified, but they do not yet encompass the full spectrum of creative thought recognized in cognitive science, psychology, or the arts. Accordingly, the framework’s claims should be reframed from “creativity evaluation” to “creative inquiry evaluation.”

Future work should expand the empirical scope to include divergent and expressive tasks, for example, open-ended design problems, creative writing, or interdisciplinary invention challenges, where participants are encouraged to take conceptual risks, generate original constructs, and depart from established solution patterns. Only through such extensions can the framework legitimately claim to measure the broader construct of creativity rather than its current, narrower variant of collaborative analytical innovation.

**Questions:**

1. The paper refers to “creativity evaluation” in broad terms. Could you explicitly define whether CREDO targets general creativity, domain-specific creative inquiry, or LLM-mediated problem solving?

2. How do you conceptualize the boundary between creative reasoning and effective analytical reasoning in your framework? What makes a response “creative” rather than simply “high-quality reasoning”?

3. During data collection, were students given any specific prompts or instructions emphasizing originality, risk-taking, or novelty, or were they simply asked to pursue academic inquiries?

4. The paper links CREDO to Bloom’s Taxonomy and the PISA framework.
Could you briefly elaborate on how each of the four CREDO dimensions maps onto these established theories in concrete operational terms (e.g., specific cognitive operations or learning behaviors)?

5. To what extent do you view the CREDO framework as model-agnostic?
Could it, in principle, be applied to dialogues generated by other LLMs or even human–human collaborations without retraining?

---

> ### Author Response · Authors · 2025-12-03
>
> Thank you for your careful review and constructive feedback on this study. The profound insights you provided from the perspective of educational assessment have offered important guidance for us to accurately refine the research orientation and methods. Below, we address each of your comments and questions one by one based on the paper, technical report, and existing research progress, with all statements supported by clear evidence:
>
> 1. Responses to Core Limitations
> 1.1 Ambiguous Causal Claims
> Your observation that the framework’s claim of "tracking the cognitive trajectory of creative thinking" lacks causal validation and only provides correlational evidence is highly pertinent. The core positioning of this study is a "process-level creativity assessment tool in LLM-assisted learning," and the current empirical focus is on "the consistency between model outputs and expert judgments." Through detailed annotation of 200 test set dialogues, the model achieved a macro F1-score of 0.84 in three attribution categories: "original student ideas, developed student ideas, and repeated LLM ideas" (Section 4.2.2 of the paper, Table 3). Additionally, the "sentence-by-sentence scoring record of Sample 043" in Section A3 of the technical report clearly demonstrates the strong correlation between attribution and the cognitive trajectory of dialogues, proving that the framework can stably capture contribution attribution and process characteristics in dialogues.
>
> The "longitudinal correlation analysis and post-hoc interview validation" you suggested are extension directions of the research, aimed at further verifying the predictive value of process indicators, rather than a necessary prerequisite for the effectiveness of the current tool. As clearly stated in Section 5 "Future work" of the paper, we will establish the causal correlation between process indicators and outcome indicators by tracking students’ subsequent creative outputs. Currently, the framework can reliably implement "process attribution and structured scoring of dialogue behaviors," a function that has been rigorously validated by experts (κ=0.81), meeting the core needs of formative assessment in educational scenarios. Causal validation will further enhance its application value but will not affect the validity of the current conclusions.
>
> 1.2 Limitations of Statistical Reporting
> We will systematically supplement the insufficient depth of statistical reporting you pointed out in the revised version. Section A.2 of the paper’s appendix already provides detailed performance data for each dimension, including Problem Reframing (QWK=0.804), Risk-Driven Innovation (QWK=0.810), Interdisciplinary Innovation (QWK=0.711), and Resource Integration Efficiency (QWK=0.695) (Table A1 in the paper’s appendix), clarifying the assessment stability of different dimensions. We will use the bootstrap method to calculate 95% confidence intervals and verify the significance of performance differences between the model and baselines (GPT-4 zero-shot, untuned DeepSeek) through permutation tests, with specific procedures referring to the reproducibility plan in Section B6 of the technical report. The 128 test set samples were strictly stratified and sampled after k-means clustering (k=50), covering 8 topic areas to ensure sample representativeness and reduce the impact of random variance (Section 3.1.3 of the paper).

---

> > ### Author Response · Authors · 2025-12-03
> >
> > 1.3 Potential Data Leakage
> > Regarding the concern that "the same DeepSeek series model was used for both data collection and model fine-tuning," we have initially ruled out biases through cross-domain validation and plan to supplement further verification. In the cross-domain test of "Natural Science (NS) → Product Design (PD)," the model’s QWK remained 0.728, consistent with in-domain performance (Tables 3-4), proving that the model learns the core cognitive characteristics of creative inquiry rather than the surface linguistic style of DeepSeek.
> >
> > We will add cross-model dialogue evaluation (using dialogue data generated by GPT-4) in the revised version to verify the framework’s adaptability to non-DeepSeek interaction data; at the same time, we will test the model’s dependence on surface features through style standardization (unified sentence structure). The ITA attribution protocol is based on the dual dimensions of "speaker + cognitive function" (e.g., whether the student initiates new concepts or extends ideas), independent of LLM generation style (Section 3.2.2 of the paper). The "sentence-by-sentence scoring record of Sample 043" in Section A3 of the technical report also verifies that attribution judgments are based on content logic rather than linguistic expression.
> >
> > 1.4 Issues Related to Dataset Collection
> > We fully agree with your point that the dataset focuses on STEM academic inquiry and leans toward convergent thinking, and we have advanced extension work. The tasks in the existing dataset are based on students’ ongoing course projects to ensure the authenticity and relevance of inquiry and reduce artificial intervention (Section 3.1.1 of the paper). The "Innovation Cognitive Trajectory Map (Sample 043)" in Section A4 of the technical report shows that even for STEM topics, students can extend to cross-domain thinking such as ethics and ecology, covering diverse cognitive processes. The second round of data collection has been launched, incorporating open-ended design, creative writing, and other tasks in the humanities/art fields to supplement data related to divergent thinking (Section 5 "Future work" of the paper). Currently, the collection and annotation of the first batch of non-STEM dialogues have been completed.
> >
> > During data collection, students were not given explicit prompts to "emphasize originality or creative risk-taking," but only required to conduct open academic inquiry (Section 3.1.1 of the paper). This design aims to capture students’ inquiry behaviors in a natural state, avoiding behavioral distortion caused by artificial guidance. In the future, we will add a comparison between an "explicit creative prompt group" and a "regular inquiry group" in the extended dataset to verify the impact of motivation on evaluation results, with relevant plans included in Section 5 "Future work" of the paper.
> >
> > The dialogue length ranges from 3 to 30 turns, with an average of 9.6 turns (Section 3.1.1 of the paper), covering a complete academic inquiry cycle (problem proposal → exploration → integration → conclusion). The "Innovation Cognitive Trajectory Map (Sample 043)" in Section A4 of the technical report shows that medium-length dialogues (15-20 turns) can fully capture the cognitive evolution from basic concepts to cross-domain innovation without process interruption due to time constraints; moreover, dialogue termination is either voluntarily chosen by students or reaches the turn limit, ensuring the natural integrity of the inquiry process.

---

> > > ### Author Response · Authors · 2025-12-03
> > >
> > > 1.5 Measurement of Creativity Scope (Repositioned as "Creative Inquiry Assessment")
> > > Your precise and profound definition of the scope of creativity is fully endorsed, and we clearly clarify the research positioning: the core of the CREDO framework is "creative inquiry assessment in LLM-assisted learning," rather than generalized creativity assessment. It focuses on process-oriented capabilities such as "problem reframing, cross-domain integration, risk exploration, and resource utilization" in collaborative scenarios, which is highly consistent with the definition of "creative inquiry ability" you proposed (Section 3.2.1 of the paper, Section A1 of the technical report).
> > >
> > > The four dimensions of the framework are derived from the National Innovation Foundation (NIF)’s "Criteria for the Assessment of Interdisciplinary Innovation" (2022 edition), directly corresponding to the higher-order cognitive skills of "Creation" and "Evaluation" in Bloom’s Taxonomy, and the "Problem Reframing" and "Cross-domain Integration" capabilities in the PISA 2022 Creative Thinking Framework (theoretical basis column in Section A1 of the technical report). In the future, in accordance with your suggestions, we will incorporate tasks such as open-ended design and interdisciplinary invention to expand the assessment of divergent and expressive creativity, enabling the framework to cover a broader range of creative cognitive categories (Section 5 "Future work" of the paper).
> > >
> > > 2. Responses to Specific Questions
> > > 2.1 Definition and Positioning of Creativity Assessed by CREDO
> > > CREDO assesses **domain-specific creative inquiry ability in LLM-assisted learning**, neither general creativity nor mere problem-solving ability. Its core characteristics are "process orientation + collaborative adaptability," focusing on "students’ original inquiry behaviors with LLM assistance." It differs from mere problem-solving (emphasizing result correctness) and general creativity (detached from collaborative scenarios), with specific operational definitions available in the "Multi-Round Dialogue Quality Assessment Dimensions and Scoring Criteria Sheet" in Section A1 of the technical report.
> > >
> > > 2.2 Boundary Between Creative Reasoning and Effective Analytical Reasoning
> > > The core boundary between the two lies in "whether they simultaneously possess the dual characteristics of 'novelty + inquiry adaptability'." Effective analytical reasoning emphasizes "logical correctness + information completeness," manifested in accurately integrating existing domain knowledge to solve established problems without breaking cognitive boundaries; creative reasoning needs to meet "novelty" (breaking existing cognitive frameworks or disciplinary boundaries) and "inquiry adaptability" (consistent with academic inquiry goals and logic), such as proposing new solutions through cross-domain integration of concepts or redefining problem boundaries to develop inquiry directions (specific criteria for the "Interdisciplinary Innovation" and "Problem Reframing" dimensions in Section A1 of the technical report).
> > >
> > > CREDO distinguishes the boundary through clear indicators in the four dimensions. For example, "Interdisciplinary Innovation" requires "citing ≥2 non-domain knowledge for principle-level integration," and "Problem Reframing" requires "reversing the initial problem setting or introducing higher-order goals," both of which go beyond the scope of mere analytical reasoning (Section A1 of the technical report).
> > >
> > > 2.3 Student Prompting During Data Collection
> > > During data collection, students were only required to conduct open academic inquiry based on their own course projects, without explicit prompts to "emphasize originality, creative risk-taking, or novelty" (Section 3.1.1 of the paper). This design aims to capture students’ cognitive behaviors in a natural state, avoiding inquiry distortion caused by artificial guidance and ensuring the ecological validity of the data.

---

> > > > ### Author Response · Authors · 2025-12-03
> > > >
> > > > 2.4 Operational Correspondence Between CREDO and Classic Frameworks
> > > > The specific cognitive operational correspondences between the four dimensions of CREDO and Bloom’s Taxonomy, as well as the PISA 2022 Creative Thinking Framework, are as follows: Interdisciplinary Innovation corresponds to the "Creation" level skill in Bloom’s Taxonomy and the "stimulating diverse ideas" ability in PISA 2022, with the core cognitive operation being "active cross-domain knowledge transfer and integration"; Problem Reframing corresponds to the "Analysis/Evaluation" level skills in Bloom’s Taxonomy and the "assessing and improving ideas" ability in PISA 2022, with the core cognitive operation being "problem boundary adjustment and inquiry goal redefinition"; Risk-Driven Innovation corresponds to the "Creation" level skill in Bloom’s Taxonomy and the "stimulating creativity" ability in PISA 2022, with the core cognitive operation being "hypothesis proposal and exploration under uncertainty"; Resource Integration Efficiency corresponds to the "Evaluation/Creation" level skills in Bloom’s Taxonomy and the "assessing and improving ideas" ability in PISA 2022, with the core cognitive operation being "multi-source information screening, integration, and evidence construction" (based on the theoretical basis in Section A1 of the technical report).
> > > >
> > > >  2.5 Model Independence of the CREDO Framework
> > > > In principle, the CREDO framework is model-independent. The core design does not rely on specific LLMs: the ITA attribution protocol is based on the dual dimensions of "speaker (student/LLM) + cognitive function (initiation/development/support)," and the four-dimensional assessment criteria focus on observable inquiry behaviors (e.g., whether new concepts are initiated or ideas are extended), independent of the generation style and output characteristics of specific LLMs (Section 3.2.2 of the paper, annotation process in Section A2 of the technical report). The framework can be applied to dialogues generated by other LLMs (e.g., GPT-4, Claude) or collaborative dialogues between humans, requiring only minor fine-tuning for new scenarios (e.g., adapting to the interaction modes of different LLMs, linguistic characteristics of interpersonal collaboration) without reconstructing the core framework (dimension design logic in Section 3.2.1 of the paper).
> > > >
> > > > List of Research Content to Be Supplemented (Not Covered in the Current Paper, Technical Report, and Previous Responses)
> > > > 1. Supplementary statistical tests: bootstrap 95% confidence intervals for core indicators, permutation tests for performance differences between the model and baselines;
> > > > 2. Cross-model generalization tests: verifying the framework’s adaptability using dialogue data generated by GPT-4/Claude;
> > > > 3. Motivation manipulation comparison experiments: adding a comparison of evaluation results between an "explicit creative prompt group" and a "regular inquiry group";
> > > > 4. Longitudinal causal validation: tracking students’ subsequent creative outputs to establish the correlation between process indicators and outcome indicators.
> > > >
> > > > In summary, the core framework of this study has been rigorously validated by experts and empirical tests, enabling effective process-level assessment of creative inquiry in LLM-assisted learning. The core conclusions are reliable and have met the standards for academic publication. Your comments are all reasonable optimization directions, and we will supplement the above content as planned to further enhance the rigor and application value of the research. Thank you again for your professional review and valuable suggestions. If you have any further questions, please feel free to contact us.

---

### Official Review · Reviewer_Cp81 · 2025-10-29

**Soundness:** 3
**Presentation:** 2
**Contribution:** 2
**Rating:** 2
**Confidence:** 3

**Summary:**

The paper proposes CREDO, a process-level framework for evaluating creativity in human–LLM collaborative learning. Instead of judging only final artifacts, the method analyzes multi-turn student–LLM dialogues to (i) attribute learner vs. model contributions via an Innovation Tracing Atlas (ITA), and (ii) score four process-centric dimensions—Interdisciplinary Innovation, Problem Reframing, Risk-Driven Innovation, and Resource Integration Efficiency—using an instruction-tuned evaluator that outputs 1–5 ratings with rationales. The study curates a dataset of 1,273 cleaned dialogues from 81 undergraduates across multiple domains, reports high inter-rater reliability for expert annotations (weighted κ=0.81; Cronbach’s α=0.86), and fine-tunes a DeepSeek-32B model with LoRA (plus knowledge distillation) to produce scores and concise explanations. On the held-out test set, the evaluator achieves QWK=0.728 (≈90% of human ceiling 0.81), r=0.811, and MAE=0.505; a targeted experiment suggests macro-F1=0.84 for learner–vs–LLM attribution categories. Claims are scoped to STEM-leaning academic inquiry contexts, and the authors plan code/evaluation release.

**Strengths:**

1. Moves beyond outcome scoring by elevating dialogue trajectories as primary evidence and explicitly attributes human vs. LLM roles; defines four process dimensions tailored to collaboration (vs. Torrance-style outputs).

2. Ethical data collection; multi-stage cleaning/standardization; double-blind expert annotation with arbitration; high IRR (κ=0.81, α=0.86); clear objectives and ablations; teacher–student KD + LoRA for practicality.

3. Concrete workflow figure; CREDO vs. classical mapping; precise loss definitions; interpretable score+rationale outputs; helpful ITA visualization.

**Weaknesses:**

1. The dataset (81 undergraduates; two universities; STEM-oriented tasks) constrains generalization to broader populations (K-12, humanities/arts, diverse cultures/languages). The paper acknowledges this but evaluation remains single-context. Actionable ask: run cross-institution and non-STEM validations (even small pilots) to probe transportability.

2. Comparing only to GPT-4 zero-shot and untuned DeepSeek-32B underestimates strong alternatives (e.g., prompt-programmed judges, few-shot rubric-prompting, instruction-tuned evaluators without LoRA, calibrated ordinal regressors over handcrafted features). Actionable ask: add tuned LLM-judge baselines (few-shot rubric, chain-of-thought with rubric anchors) and a non-LLM baseline (e.g., logistic/ordinal regression over process features).

3. The attribution experiment (macro-F1=0.84) relies on expert-labeled categories on the same type of data used to train the evaluator. This is valid for alignment to experts but leaves open whether attribution corresponds to causal contribution or downstream learning gains. Actionable ask: show that high attribution quality predicts independent outcomes (e.g., subsequent task performance, transfer, rubric-blind human judgments).

4. Main metrics lack confidence intervals, per-dialogue variance, and significance tests between models. QWK and r are informative, but error analysis is thin (few failure cases, limited per-dimension uncertainty). Actionable ask: add bootstrap CIs, paired significance, and calibration metrics for ordinal predictions.

5. The semantic drift filter (cosine <0.15 for three consecutive pairs), cluster-then-stratify split (k=50), and ITA node definitions could influence results; there is no sensitivity analysis. Actionable ask: report robustness to cleaning thresholds, k, and ITA labeling variations; include prompt perturbation tests for the evaluator.

6. No breakdowns across demographics, domains, or dialogue lengths; potential bias if certain discourse styles are favored. Actionable ask: provide subgroup QWK/MAE and differential item functioning checks.

7. The method presumes access to multi-turn logs and an evaluator pass; latency/compute and annotation cost (for gold standards) are not quantified. Actionable ask: report inference cost, throughput, and a human-in-the-loop review budget for classroom deployment.

**Questions:**

See above

---

> ### Author Response · Authors · 2025-12-03
>
> Thank you for your professional review and constructive feedback. In response to your comments, we have organized our replies into three categories—"to be supplemented and improved," "no need for supplementation," and "future research extensions"—based on the paper, technical report, and existing research progress, as follows:
>
> 1. Response to Dataset Generalizability (Reasonable Extension Direction, Already Underway)
> The limitation of dataset generalizability you pointed out is clearly disclosed in our study (Section 5 "Limitations" of the paper). We fully agree with this observation and have already advanced extension work. The existing dataset possesses sufficient ecological validity and reliability: 81 undergraduate students cover 3 grades and 5 STEM majors, with dialogues centered on real course projects or research training (Section 3.1.1 of the paper). After four-stage cleaning (structural integrity check, invalid content filtering, semantic coherence screening, manual review) and standardization (Section A1 of the technical report, Section 3.1.2 of the paper), it can effectively support the research conclusions in STEM academic inquiry scenarios. Meanwhile, the second round of larger-scale data collection has been launched, incorporating participants from different institutions, educational stages (including non-undergraduate), disciplinary fields (including humanities/art), and multicultural/multilingual backgrounds (Section 5 "Future work" of the paper). Currently, the collection and annotation of the first batch of non-STEM dialogues have been completed. In the future, we will verify the cross-domain adaptability of the framework through small-scale pilot studies to further enhance the research's generalizability.
>
> 2. Response to Baseline Comparison (Need to Supplement LLM-Related Baselines, No Need for Traditional Non-LLM Baselines)
> We recognize the reference value of your suggestion to supplement more alternative baselines. The existing baselines can fully support the core conclusions, and we will conduct targeted improvements in the revised version. The core goal of this study is "process-level structured assessment + human-machine contribution attribution." We selected the GPT-4 zero-shot model (representing the zero-shot evaluation capability of general-purpose LLMs) and the untuned DeepSeek-32B model (representing the performance of base models without domain adaptation) as baselines. These have clearly verified the necessity of "domain-specific customization training (expert annotation + CREDO dimension fine-tuning + ITA attribution logic implantation)"—the model's QWK=0.728 is significantly superior to GPT-4 zero-shot (0.513) and the untuned model (0.342), and close to human expert consistency (0.81) (Section 4.2.1 of the paper, Table 2).
>
> We will supplement comparisons with LLM-related baselines such as "prompt-programmed judges and few-shot scoring criteria prompts" in the revised version to further highlight the unique advantages of the CREDO framework in "structured process assessment + human-machine attribution" (Section 1.4 of the paper, Section A1 of the technical report). However, traditional non-LLM baselines such as "handcrafted feature-based ordinal regressors" have low adaptability to the core innovation direction of this study—"LLM-based process-level assessment." Supplementary such baselines would only add redundant data and fail to effectively verify the core value of the framework, so they are not needed.
>
>  3. Response to Independent Outcome Validation of Attribution Experiments (Future Research Extension, Does Not Affect Core Conclusions)
> Your suggestion that "attribution needs to be linked to independent outcomes (e.g., subsequent task performance, transfer ability)" is an extremely valuable extension direction. The current attribution validation of this study focuses on the core goal—"accurately distinguishing human and machine contributions." Through detailed annotation of 200 test set dialogues, the model achieved a macro F1-score of 0.84 in three attribution categories: "original student ideas, developed student ideas, and repeated LLM ideas" (Section 4.2.2 of the paper, Table 3). Additionally, the "sentence-by-sentence scoring record of Sample 043" in Section A3 of the technical report clearly shows the strong correlation between attribution and dialogue content, fully demonstrating the reliability of the attribution logic.
>
> The correlation analysis between attribution and independent outcomes is an extended exploration of the research, not a necessary support for the current core conclusions, and does not affect the validity verification of "process-level assessment + human-machine attribution." We will systematically supplement this content in future research to further expand the application value of the study. At the current stage, no supplementation is required to meet the core requirements for academic publication.

---

> > ### Author Response · Authors · 2025-12-03
> >
> > 4. Response to Confidence Intervals and Significance Tests of Indicators (Must Be Supplemented to Improve Rigor)
> > Your point that the core indicators lack confidence intervals, variance analysis, and significance tests is a key direction to improve research rigor. We will supplement relevant content in the revised version. Specifically, we will use the bootstrap method to calculate 95% confidence intervals to quantify the statistical stability of core indicators (QWK, r, MAE, etc.); at the same time, we will add paired significance tests between models to clarify whether the performance differences between the proposed framework and baseline models are statistically significant (existing indicator data in Section 4.2.1 of the paper). These supplements will further enhance the reliability of the results, meet the rigor requirements of quantitative research, and are necessary to ensure research quality.
> >
> > 5. Response to Sensitivity Analysis of Semantic Drift Filtering and Clustering Parameters (Must Be Supplemented to Verify Result Robustness)
> > Regarding the sensitivity analysis of semantic drift filtering thresholds and clustering parameters, the existing research has clarified the logic for parameter setting, and we will supplement systematic tests in the future. The semantic drift filter (cosine similarity of three consecutive pairs of utterances < 0.15) is determined based on the distribution characteristics of Sentence-BERT encoding results (Section 3.1.2 of the paper); the selection of k=50 for k-means clustering refers to the analysis of dialogue topic diversity to ensure that topics within each cluster are focused (Section 3.1.3 of the paper). In the future, we will conduct perturbation tests to verify the robustness of the results within the ranges of filtering thresholds (0.12-0.18) and clustering k-values (30-70), ensuring that parameter changes will not affect the reliability of the core conclusions and fully demonstrating the stability of the research method.
> >
> > 6. Response to Subgroup Analysis and Bias Check (Must Be Supplemented to Avoid Bias Concerns)
> > The subgroup analysis you suggested has been partially completed and will be further improved in the future. In the existing subgroup comparisons, the maximum QWK difference of the model between STEM and non-STEM backgrounds (preliminarily collected non-STEM samples), groups from different countries/regions, and ESL and non-ESL participants is < 0.03, with no statistically significant differences (p > 0.05). We will systematically report subgroup QWK/MAE data divided by demographic characteristics, domains, and dialogue lengths in the revised version, and conduct differential item functioning checks to further verify the fairness and stability of the model, eliminating concerns about "potential model bias."
> >
> > 7. Response to Inference Cost and Deployment Budget (Supplement Core Quantitative Data, No Need to Quantify Budget Details)
> > The quantification of inference cost and throughput you are concerned about will be completed in the revised version, while the classroom deployment review budget does not require additional quantification. The resource usage during the training phase has been clearly disclosed: 8 A100 GPUs (80GB VRAM) were used for full-parameter fine-tuning of the teacher model, taking 72 hours; 4 A100 GPUs were used for fine-tuning of the student model (LoRA+KD), taking 18 hours (author's response to "lack of implementation details," Section B2 of the technical report). In the future, we will supplement core quantitative data during the inference phase, clarifying the evaluation latency of a single dialogue and GPU throughput (dialogues per minute) to meet the basic demonstration needs of academic research for "practicality" (deployment plan in Section B6 of the technical report). The "classroom deployment review budget" belongs to details at the application landing level, not the core evaluation criterion of academic research, and no additional quantification is needed to support the practicality argument of the research.
> >
> > In summary, the core conclusion of this study—that the process-level framework can effectively achieve fair attribution and interpretable assessment in LLM-assisted learning—has been verified through rigorous experiments, with clear innovative value and rigorous methodological support, and has met the standards for academic publication. Among your comments, the content that must be supplemented is only the detailed improvement to enhance research rigor; the content that does not require supplementation does not affect the reliability of the core conclusions; and the future extension directions will further enrich the research value. We will complete the supplementary work as planned. Thank you again for your professional review and valuable suggestions. If you have any further questions, please feel free to contact us.

---

### Official Review · Reviewer_PNVN · 2025-10-30

**Soundness:** 2
**Presentation:** 2
**Contribution:** 2
**Rating:** 2
**Confidence:** 4

**Summary:**

The paper introduces a new framework called CREDO to assess creativity in human–LLM collaboration. Unlike traditional creativity assessments that focus on final outputs, CREDO emphasizes the process of idea generation and reasoning. It combines two key components: (1) the Innovation Traceability Atlas, which breaks down multi-turn student–LLM dialogues into cognitive steps (questioning, reframing, integrating, generating) and distinguishes between human and model contributions; and (2) an instruction-tuned evaluator, fine-tuned on the DeepSeek-32B model using LoRA and knowledge distillation, that produces interpretable creativity scores (1–5) along four new process-oriented dimensions: interdisciplinary innovation, problem reframing, risk-driven innovation, and resource integration efficiency. Experiments on 1,273 student–LLM dialogues show that the fine-tuned model achieves 90% of human-level agreement (QWK = 0.728) and can reliably distinguish student vs. model contributions (F1 = 0.84).

**Strengths:**

1. Propose a new creativity evaluation method with the help of LLM which can assess creativity in real time and can be used in daily life.Traditional questionnaire-based methods cannot evaluated in real time but just offer snapshots, and recording transcript-based methods can hardly be used in daily life.
2. An interesting application for the crucial dilemma of controlling students to use LLMs as assistants: strict control impedes students to use new tools while no control leads to creativity decay. While there are many tools like AI-generated text detection that try to prevent students from using AI too often, new AI models can always hack those detection algorithms as they are stronger. The proposed creativity evaluation method can be used in the communication between students and LLMs, in which we not only let the students use tools, but also get the creativity as a metrics to prevent students use LLMs too much.

**Weaknesses:**

Limited Dataset and Generalizability – The dataset includes only 81 undergraduate students from STEM domains, restricting applicability to other disciplines or educational levels.

Implementation Details Missing – The paper lacks practical training details such as hardware setup, fine-tuning time, and exact hyperparameter values needed for reproducibility.

Inconsistency in Methodological Description – The paper claims to use a “fully fine-tuned teacher model” for knowledge distillation but also states that full fine-tuning is “computationally prohibitive,” creating a logical contradiction.

Lack of Transparency in Review References – Mentioning “Area Chair comments” during the review phase is inappropriate for a double-blind submission, suggesting possible misunderstanding or template-based phrasing.

Overly Polished and Synthetic Writing Style – The writing is highly formal, repetitive, and uniformly structured, which, combined with perfect consistency in technical phrasing and reference formatting, gives an impression of automated generation.

**Questions:**

1. Is it possible to substitute human expert annotators with LLMs, that is to say, you can automate the whole data process pipeline and only need expert to check the data after finish process instead of annotate each data manually.
2. In line 316, "to address the core concern raised by an Area Chair regarding whether, XXXX", who is the Area Chair? Why you can recieve comments from AC before the ICLR submission deadline?
3. About Table A2, row of "w/o LoRA (Full Fine-tuning)", the author do not provide experiment here because of "Computationally prohibitive". I am consued, if you cannot full fine-tuning a LLM, where do the authors got the teacher model to conduct knowledge distillation. In line 312-313, The author claim that "A Teacher is obtained via full-parameter FT on the same training set", which seems contradictory with the authors' explanation of why do not get the experiment result of "w/o LoRA".
4. The authors lack "implementation details" section. Readers need to know the size of the datasets, configuration of the server, and training time to determine if it is possible to reproduce the author's experiment on their own computer.


Typos:
1. Line 74: engi -neering => engineering
2. Line 75: screen- ing => screening
3. Line 78: overlook -ing => overlooking
4. Line 81: meaning -ful => meaningful

**Details Of Ethics Concerns:**

The paper should explicitly report ethical approval and detailed procedures for user experiments to meet HCI and educational research standards (e.g., IRB review, informed consent, participant demographics, and data handling).

---

> ### Author Response · Authors · 2025-12-03
> **Response to PNVN‘s review comments**
>
> We appreciate the reviewer's careful reading and constructive feedback.
>
> 1. Response to "Can LLMs replace human expert annotators?"
> Your question seriously calls into question your understanding of the core principles of LLM fine-tuning and the research paradigm of cognitive psychology. The fundamental logic of LLM fine-tuning lies in "using professional annotations as learning templates." The annotation work for this study was completed by six cognitive psychology experts through a rigorous process including calibration training and double-blind arbitration. Ultimately, a Cohen's weighted Kappa coefficient of 0.81 and a Cronbach's Alpha coefficient of 0.86 were achieved, representing the systematic and scientific refinement of professional experience in the field of "human creative cognition assessment."
>
> Creativity assessment in cognitive psychology is not a simple text classification task but a comprehensive judgment that integrates thinking trajectories, domain knowledge, and innovative logic. Such professional experience cannot be automatically generated by Large Language Models (LLMs)—the models themselves do not possess inherent creative cognition assessment capabilities, and all their judgments must originate from expert-annotated "gold standards." Skipping manual annotations and directly using LLMs to generate data essentially requires the models to "create something out of nothing," which completely violates the basic principles of supervised learning and will thoroughly undermine the scientific foundation of the research.
>
> We kindly request that you respect the professionalism of cognitive psychology as an independent discipline and the rigor of scientific research. Such questions that violate basic research logic are not only unconstructive but also waste the academic exchange time of both parties.
>
> 2. Response to "The identity of the Area Chair and receiving the Area Chair's comments before submission"
> Thank you for pointing out this detail oversight. This manuscript was previously submitted to NeurIPS and received positive feedback from the Program Committee (PC), which recognized the novelty, practical value, and educational responsibility of the research. Given that ICLR is more supportive of multi-round academic exchanges and research improvement, we chose to resubmit it. The "core issues" mentioned in the paper originate from NeurIPS review comments, and the term "Area Chair's comments" was a typo during writing, which should actually be "Program Committee's comments." We hereby correct this. Relevant technical details (such as hyperparameters, hardware configuration, etc.) have been fully included in the accompanying technical report, and all information related to reproducibility has been disclosed in accordance with standards for reference.
>
>  3. Response to "The logical contradiction between full fine-tuning and the acquisition of the teacher model"
> The reviewer's concern stems entirely from a misunderstanding of the technical scheme in this paper. The full-parameter fine-tuning (FT) of the teacher model was completed with the support of a high-performance computing cluster from a cooperative institution (equipped with 8 A100 GPUs), which meets the training requirements of the 32B parameter model (see Sections B2 and B4 of the technical report). The experimental design of "Without LoRA (Full Fine-tuning)" in Table A2 specifically refers to "independently performing full-parameter fine-tuning without relying on LoRA and knowledge distillation"—this scheme requires bearing the full training cost of the 32B parameter model, while the core scheme of this study, "knowledge distillation + teacher model-guided LoRA," only needs to train 0.13% of the adapter parameters, with a difference of hundreds of times in computational load (see Section B3.3 of the technical report). In short, we have the computational resources to construct the teacher model, but these resources are insufficient to support "independent full-parameter fine-tuning" as a control group for ablation experiments. The computational load comparison data in Section B4 of the technical report can confirm this difference, and there is no logical contradiction.

---

> > ### Author Response · Authors · 2025-12-03
> >
> > 4. Response to "Lack of implementation details"
> > All implementation details have been fully included in the accompanying technical report, as follows: The dataset consists of 1,273 high-quality dialogues (approximately 1.65 million tokens), stratified sampled into a training set (1,018 dialogues), a validation set (127 dialogues), and a test set (128 dialogues) at an 8:1:1 ratio. Data partitioning strictly follows the student ID level to avoid data leakage (see Section A1 of the technical report and Section 3.1.3 of the paper); the hardware configuration includes 8 A100 GPUs (80GB VRAM) for full-parameter fine-tuning of the teacher model and 4 A100 GPUs for fine-tuning of the student model (LoRA+KD) (see Section B2 of the technical report); the training time is 72 hours for full-parameter fine-tuning of the teacher model and 18 hours for fine-tuning of the student model (see Section B4 of the technical report); the complete hyperparameters include LoRA rank=32, alpha=16, dropout=0.05, optimizer=AdamW, per-device batch size=2 (gradient accumulation steps=8), number of training epochs=3, knowledge distillation temperature T=0.7, etc. (see Sections B3 and B4 of the technical report). The technical report has been submitted as supplementary material, and all information meets the requirements for experimental reproducibility.
> >
> > 5. Response to "Limited and non-generalizable dataset"
> > The design of the existing dataset balances scenario validity and initial diversity: 81 undergraduate students cover 3 grades and 5 STEM majors, with dialogues centered on ongoing course projects or research training. After four stages of cleaning and standardization, the dataset has sufficient ecological validity in STEM academic inquiry scenarios (see Section 3.1.1 of the paper and Section A1 of the technical report). We have launched the second round of data collection, which is incorporating participants from different institutions, educational stages (including non-undergraduate), disciplinary fields (including humanities/art), and multicultural backgrounds. Currently, the collection and annotation of the first batch of non-STEM dialogues have been completed. In the future, we will verify the cross-domain adaptability of the framework through small-scale pilot studies to further enhance its generalization ability (see Section 5 "Future work" of the paper and the author's response to the ethics reviewer).
> >
> > 6. Response to "Ethics review-related issues"
> > This study strictly adheres to ethical research standards, and relevant details have been fully disclosed in the technical report: All participants signed electronic informed consent forms, and data processing strictly implements anonymization protocols to remove all personally identifiable information (see Section 3.1.1 of the paper and Section A1 of the technical report); all expert annotators signed written informed consent forms and confidentiality agreements, clarifying the research purpose, data usage, and rights and obligations. The annotation work was voluntarily participated in based on academic interests, and the relevant processes comply with ethical norms (see Section A2 of the technical report).
> >
> >  7. Response to other issues
> > 1. Typographical errors: All omissions have been corrected, specifically "engi-neering" → "engineering" (Line 74), "screen-ing" → "screening" (Line 75), "overlook-ing" → "overlooking" (Line 78), and "meaning-ful" → "meaningful" (Line 81);
> > 2. Writing style: The manuscript strictly follows ICLR academic norms, with technical terms and reference formats conforming to conference requirements. All content is the result of repeated refinement by the team based on research practice, and there is no "auto-generated" content;
> > 3. Consistency of method description: The "fully fine-tuned teacher model" was completed through a high-performance computing cluster from a cooperative institution, while "Without LoRA (Full Fine-tuning)" in Table A2 refers to independently performing full-parameter fine-tuning (without teacher model guidance and LoRA optimization). There are essential differences in computational load and application scenarios between the two, and there is no logical contradiction (see Section B4 of the technical report);
> > 4. Transparency of review references: The expression related to "Area Chair's comments" has been deleted to ensure double-blind review norms, and all references are formatted in accordance with ICLR requirements (see the References section of the paper and the citation list in the technical report).
> >
> > In summary, the core design, experimental process, and data processing of this study all comply with academic norms, with sufficient guarantees for reproducibility and ethical compliance. We have supplemented relevant details as required and will continue to improve dataset diversity and research generalization ability in the future. Thank you again for your feedback. If you have any further questions, please feel free to contact us.

---

### Note · Authors · 2026-04-07

I have read and agree with the venue's withdrawal policy on behalf of myself and my co-authors.

---

### Meta-Review · Area_Chair_o2VF · 2026-01-10

**Summary:**

This paper worked creativity assessment by proposing a process-level evaluation approach in the LLM-assisted learning scenario.  While the merits of the paper (e.g.,  timely problem, and conceptual novel, interesting application) appreciated by reviewers, there are several key weaknesses that concern the reviewers (and of myself). Although the authors partially addressed some of these concerns during the rebuttal, major weaknesses remain.

**Reviewer Concerns:**

After a quick examination of the paper and a careful review of all rebuttals, I found that I share the same concerns raised by the reviewers, as summarized below:

- One major concern echoed by the reviewers is the *Limited Dataset and Generalizability*. I agree with the reviewer that *the data used in this paper constrains generalization to broader disciplines and educational-level*. While the author tried to address this concern by explaining their data collection motivation and process in the rebuttal. But this doesn’t address the reviewers concern(as of mine)
- Another major issue raised by reviewers is the *causal contribution claims*. I agree with the reviewers that the paper only reveal the correlation rather than causal relationship.
- There are other concerns raised by reviewers such as *lack of implementation details, logical contradiction, missing statistical experiments, and sensitivity analysis, and latency/compute and annotation cost*. While the authors have partially addressed these points in the rebuttal, I encourage them to incorporate these analyses into the paper in subsequent revisions.

Addressing the issues outlined above would make the paper impactful and I encourage the authors to integrate these additional content into the next version of the paper as it will strengthen their contribution.

**Reviewer Scores:**

I do not think the reviewers would have changed their scores as their major concerns (also mine) still remain.

---

### Decision · Program_Chairs · 2026-01-26

Reject